# Antitumor Effects of Intravenous Natural Killer Cell Infusion in an Orthotopic Glioblastoma Xenograft Murine Model and Gene Expression Profile Analysis

**DOI:** 10.3390/ijms25042435

**Published:** 2024-02-19

**Authors:** Takayuki Morimoto, Tsutomu Nakazawa, Ryosuke Matsuda, Ryosuke Maeoka, Fumihiko Nishimura, Mitsutoshi Nakamura, Shuichi Yamada, Young-Soo Park, Takahiro Tsujimura, Ichiro Nakagawa

**Affiliations:** 1Department of Neurosurgery, Nara Medical University, Kashihara 634-8521, Nara, Japan; t.morimoto@naramed-u.ac.jp (T.M.); nakazawa@naramed-u.ac.jp (T.N.); r.maeoka@naramed-u.ac.jp (R.M.); fnishi@naramed-u.ac.jp (F.N.); mnaka@grandsoul.co.jp (M.N.); syamada@naramed-u.ac.jp (S.Y.); park-y-s@naramed-u.ac.jp (Y.-S.P.); nakagawa@naramed-u.ac.jp (I.N.); 2Department of Neurosurgery, Nara City Hospital, Nara 630-8305, Nara, Japan; 3Grandsoul Research Institute for Immunology, Inc., Uda 633-2221, Nara, Japan; takahiro@grandsoul.co.jp

**Keywords:** glioblastoma, NK cell, murine orthotopic xenograft model

## Abstract

Despite standard multimodality treatment, containing maximum safety resection, temozolomide, radiotherapy, and a tumor-treating field, patients with glioblastoma (GBM) present with a dismal prognosis. Natural killer cell (NKC)-based immunotherapy would play a critical role in GBM treatment. We have previously reported highly activated and ex vivo expanded NK cells derived from human peripheral blood, which exhibited anti-tumor effect against GBM cells. Here, we performed preclinical evaluation of the NK cells using an in vivo orthotopic xenograft model, the U87MG cell-derived brain tumor in NOD/Shi-scid, IL-2RɤKO (NOG) mouse. In the orthotopic xenograft model, the retro-orbital venous injection of NK cells prolonged overall survival of the NOG mouse, indirectly indicating the growth-inhibition effect of NK cells. In addition, we comprehensively summarized the differentially expressed genes, especially focusing on the expression of the NKC-activating receptors’ ligands, inhibitory receptors’ ligands, chemokines, and chemokine receptors, between murine brain tumor treated with NKCs and with no agents, by using microarray. Furthermore, we also performed differentially expressed gene analysis between an internal and external brain tumor in the orthotopic xenograft model. Our findings could provide pivotal information for the NK-cell-based immunotherapy for patients with GBM.

## 1. Introduction

Glioblastoma (GBM), the most prevalent and aggressive primary brain tumor, is associated with a dismal prognosis and poor performance status. The standard treatment involves maximal safe resection with monitoring, navigation, or awake surgery, followed by radiation treatment and adjuvant temozolomide [1,2]. This established protocol is known as the “Stupp regimen”. Tumor-treating fields have recently become a standard treatment modality for GBM. Despite multimodal interventions, patients with GBM typically demonstrate a median overall survival of only 20.5 months [3].

Several novel strategies have been explored for GBM therapy. G47∆, a triple-mutated, third-generation oncolytic herpes simplex virus type 1, has been approved as the first oncolytic virus product in Japan, and its administration is reportedly associated with a prolonged median overall survival of 20.2 months after G47∆ initiation [4]. Although checkpoint inhibitors are extensively used in other cancer treatment [5,6], most clinical trials have not demonstrated their efficacy against GBM [7,8,9]. Cloughesy et al. reported that in comparison to adjuvant administration of pembrolizumab against recurrent GBM, preoperative administration significantly extended overall survival [10]. However, their investigation involved a small sample size of only 35 patients with recurrent GBM, necessitating further investigations.

GBM is an immunologically “cold” tumor, with the tumor microenvironment (TME) mainly comprising tumor-associated macrophages with fewer T cells and natural killer cells (NKCs) [11,12,13]. This highly immunosuppressive TME influences brain tumor outgrowth and induces resistance against immunotherapy. Considering the limited efficacy of immunotherapies primarily based on T cell activation, we focused on NKC-based immunotherapy, which offers distinct advantages compared to T-cell-based immunotherapy. NKCs, discovered over 40 years ago, are innate lymphocytes that play an important role in controlling microbial infections and tumor progression [14,15,16]. NKC-based treatments have several advantages in cancer immunotherapy. First, NKCs are able to recognize cancer cells using a balance of multiple activating and inhibitory receptors without being limited by the expression of major histocompatibility complex (MHC) class I [14,17]. On the other hand, T cells identify tumor cells via T cell receptors, which recognize fragments of a single antigen as peptides bound to MHC molecule [18]. The ability of NKCs to recognize MHC class I-deficient tumor cells, which is also called “missing-self” recognition, provides an advantage in eliminating cancer cells evading immune responses [19]. Second, activated NKCs play a key role in recruiting conventional type 1 dendritic cells and subsequently CD8^+^ T cells, promoting the cancer immunity cycle by engaging with other immune components [20,21]. Third, unlike T cells, NKCs do not induce graft-versus-host disease because they lack T cell receptors [22,23,24]. These advantages of NKCs could potentially transform the GBM TME from a “cold” to “hot” tumor. In clinical trials, the efficacy of adoptive NKC therapy against recurrent glioma was investigated in a phase 2 trial, and the patients treated with lymphokine-activated killer (LAK) cells and IL-2 to the CNS exhibited higher survival rate [25,26]. Ishikawa et al. also reported that NKC therapy for recurrent malignant glioma was safe and partially effective [27]. These results indicate that the addition of NKCs in the GBM TME, where NKCs are absent, would be effective to treat the patients with GBM. However, these studies utilized low-purified NKCs to treat GBM.

We previously reported a unique technique to derive ex vivo-expanded highly purified NKCs from human peripheral blood mononuclear cells using a chemically defined and feeder-free method [28]. These NKCs exhibited high activity against two-dimensional-culture and three-dimensional-spheroid models derived from GBM cell lines [29]. Furthermore, we demonstrated the antitumor activity of NKCs using a xenograft model of subcutaneously implanted U87MG cells in nonobese diabetes/severe combined immunodeficiency/IL2rγ null (NOG) mice [30]. Herein, we investigated the antitumor activity of NKCs cultured using our unique method in an orthotopic GBM xenograft model in NOG mice. We also performed gene expression and enrichment analyses using microarray data from tumors in the xenograft model with or without NKC treatment. We believe that our findings provide crucial information for clinical trials assessing the efficacy of NKC-based immunotherapy in patients with GBM.

## 2. Results

### 2.1. Cytotoxicity-Mediated Growth Inhibition Assay

We investigated the growth inhibitory effects of human primary NKCs on two GBM cell lines using the RTCA system. After culturing U87MG and T98G cells for 1 day, NKCs cultured for 14 days were added to each well at effector-to-target cell ratios of 1:1 and 1:2. The growth inhibitory effect mediated by cytotoxicity was clearly detected (Figure 1a,b). NKCs were observed to significantly inhibit the growth of both GBM cell lines.

### 2.2. In Vivo Orthotopic Xenograft Assays and Histological Analysis

We examined the in vivo therapeutic efficacy of allogeneic NKCs cultured using our specific method against GBM cell lines using an orthotopic xenograft murine model. To investigate the pure anti-activity of NKCs and establish easily the orthotopic brain tumor, we utilized NOG mice in this assay. U87MG cells (10^5^ cells) were implanted into the brain of NOG mice (Figure 2a). In all groups, treatment agents or control IL-2 were injected via the retro-orbital sinus. Further, different NKCs derived from two healthy volunteers were examined. In comparison with the negative-background group, NKC-treated groups were significantly associated with a longer survival time (Figure 2b). However, there was no significant difference between the intravenous once- and twice-infusion groups.

Histological analyses revealed that tumors detected in all groups exhibited similar histological features to human GBM (Figure 2c).

### 2.3. Gene Expression and Enrichment Analyses

Gene expression profiles of intracranial tumors from the orthotopic xenograft model treated with or without NKCs were assessed using microarray (Figure 3a). We focused on glioma stem cell (GSC) markers, extracellular marker (ECM), chemokines, chemokine receptors, NKC-activating receptor ligands, and NKC inhibitory receptor ligands, as stated earlier (Appendix A). On NKC treatment, the expression of the GSC markers SOX2 and MSI1 was upregulated in intracranial tumors (fold change: 2.01 and 2.39, respectively); DABG values for the NKC-treated group were 11.65 and 6.84 and those for the negative-background group were 10.65 and 5.58, respectively (Figure 3b). Further, the expression of MYC, CD44, and STAT3 was downregulated in the NKC-treated group (fold change: −2.25, −3.21, and −2.39, respectively); DABG values for the NKC-treated group were 5.17, 8.87, and 8.09 and those for the negative-background group were 6.35, 10.55, and 9.35, respectively. Among the ECM markers, the expression of LAMA1 was upregulated; the DABG value for the NKC-treated group was 5.48 and that for the negative-background group was 4.41 (Figure 3c). The expression of SNED1, FN1, and COL6A1 was downregulated in the NKC-treated group (fold change: −5.12, −4.42, and −8.29, respectively); DABG values for the NKC-treated group were 8.60, 9.59, and 8.98 and those for the negative-background group were 10.96, 11.73, and 12.03, respectively. Among the chemokines, the expression of CXCL14, CCL13, CCL11, and CCL19 was upregulated in the NKC-treated group (fold change: 8.19, 2.33, 2.12, and 2.09, respectively); DABG values for the NKC-treated group were 8.1, 6.94, 5.49, and 7.7 and those for the negative-background group were 5.07, 5.72, 4.4, and 6.64, respectively (Figure 3d). Among the chemokine receptors, the expression of CCR5 was upregulated in the NKC-treated group (fold change: 1.79); the DABG value for the NKC-treated group was 5.59 and that for the negative-background group was 4.75 (Figure 3e). The expression status of CCR5 changed from False to True, as indicated by the DABG value for the NKC-treated group. Among the NKC-activating receptor ligands, the expression of NID1 and PDGFD was upregulated in the NKC-treated group (fold change: 3.35 and 2.27, respectively); DABG values for the NKC-treated group were 7.47 and 5.19 and those for the negative-background group were 5.73 and 4.01, respectively (Figure 3f). The expression of CLEC2B and CD70 was downregulated in the NKC-treated group (fold change: −2.32 and −5.64, respectively); DABG values for the NKC-treated group were 5.48 and 5.08 and those for the negative-background group were 6.69 and 7.57, respectively. Among the NKC inhibitory receptor ligands, the expression of CDH2 was upregulated and that of HLA-E and PTDSS1 was downregulated in the NKC-treated group (fold change: 2.1, −3.38, and −2.55, respectively); DABG values for the NKC-treated group were 9.64, 5.55, and 8.27 and those for the negative-background group were 8.57, 7.31, and 9.61, respectively (Figure 3g).

According to enrichment analysis, relative to the negative-background group, the following gene sets were downregulated in intracranial tumors treated with NKCs: postsynaptic specialization membrane, postsynaptic density membrane, transmitter-gated-channel activity, glutamate-receptor activity, and postsynaptic membrane (Figure 3h,i).

### 2.4. Gene Expression Analysis of the External and Internal Layers of Intracranial Tumors

We examined the gene expression patterns between the external and internal layers of intracranial tumors from the orthotopic xenograft mouse model treated with NKCs (Figure 4a, Appendix A). In the external layer, NKC treatment upregulated the expression of the GSC markers CDH5 and MSI1 in intracranial tumors (fold change: 2.42 and 2.24, respectively); DABG values for the external layer were 5.83 and 7.67 and those for the internal layer were 4.56 and 6.51, respectively (Figure 4b). Further, the expression of NES and L1CAM was downregulated in the external layer (fold change: −2.18 and −2.28, respectively); DABG values for the external layer were 6.55 and 8.58 and those for the internal layer were 7.74 and 9.7, respectively. Among the ECM markers, the expression of SNED1, LTBP1, COL4A6, and CDH1 was upregulated (fold change: 6.82, 3.48, 2.80, and 2.79, respectively); DABG values for the external layer were 10.14, 10.9, 5.71, and 4.98 and those for the internal layer were 7.37, 9.10, 4.22, and 3.50, respectively (Figure 4c). The expression of LUM and MMP16 was downregulated in the external layer (fold change: −2.11 and −4.50, respectively); DABG values for the external layer were 7.52 and 7.64 and those for the internal layer were 8.61 and 9.81, respectively. Among the chemokines, the expression of CCL24, CCL13, and CCL27 was upregulated in the external layer (fold change: 5.23, 2.64, and 2.32, respectively); DABG values for the external layer were 6.86, 7.49, and 9.55 and those for the internal layer were 4.48, 2.64, and 2.32, respectively (Figure 4d). The expression of CXCL14, CCL19, CCL25, CCL2, and CX3CL1 was downregulated in the external layer (fold change: −6.06, −2.41, −2.36, −2.08, and −2.02, respectively); DABG values for the external layer were 10.94, 6.59, 5.04, 5.97, and 4.7 and those for the internal layer were 13.54, 7.86, 6.27, 7.03, and 5.71, respectively. Among the chemokine receptors, the expression of CCR2 was upregulated in the external layer (fold change: 2.04); the DABG value for the external layer was 6.79 and that for the internal layer was 5.76 (Figure 4e). The expression of CXCR3 was downregulated in the external layer (fold change: −2.57); the DABG value for the external layer was 6.25 and that for the internal layer was 7.61. Among the NKC-activating receptor ligands, the expression of TNFSF4 and NCR3LG1 was upregulated in the external layer (fold change: 2.06 and 2.00, respectively); DABG values for the external layer were 4.72 and 7.12 and those for the internal layer were 3.68 and 6.12, respectively (Figure 4f). The expression of NID1 and CD70 was downregulated in the external layer (fold change: −4.83 and −3.92, respectively); DABG values for the external layer were 10.45 and 5.46 and those for the internal layer were 12.72 and 7.44, respectively. Among the NKC inhibitory receptor ligands, the expression of CDH1 was upregulated and that of CDH2 was downregulated in the external layer (fold change: 2.79 and −4.73, respectively); DABG values for the external layer were 4.98 and 12.27 and those for the internal layer were 3.50 and 14.52, respectively (Figure 4f).

## 3. Discussion

While numerous effective adoptive immunotherapies have been reported against various cancers [5,6,31], there is currently no efficient immunotherapy specifically tailored for GBM treatment. Existing immunotherapies, including immune-checkpoint inhibitors or chimeric antigen receptor T-cell therapies, primarily focus on T cell activation. In light of this, we directed attention to NKC-based immunotherapy, which presents several advantages over T-cell-based approaches. NKC-based immunotherapy offers the capability of recognizing multiple antigens or cancer cells, particularly those with downregulated expression of MHC class 1 molecule. This is pivotal for addressing the intra- and intertumoral heterogeneity exhibited by GBM [32,33]. Intratumoral heterogeneity refers to hypoxia, stem cells, resistance regions, transformed neuronal regions, proliferative regions, and mutation sites within GBM tissues [33]. Notably, the pattern of intratumoral heterogeneity in recurrent GBM after adjuvant treatment differs from that in primary GBM [32]. Tackling such molecular intricacies is challenging, with therapies targeting a single antigen or molecule, a limitation associated with T-cell-based immunotherapy or molecular target drugs. In contrast, NKC-based immunotherapy has the potential to overcome intratumoral heterogeneity by leveraging its tumor recognition system involving multiple activating and inhibitory receptors. Herein, our findings demonstrated the significant antitumor effects of NKCs in vitro and in vivo, offering a valuable advantage for GBM treatment. Activated NKCs were found to recruit other immune components, such as conventional type 1 dendritic cells and CD8^+^ T cells. Notably, in the GBM TME, where NKCs are reportedly absent [13,34], NKC administration can markedly transform the immunosuppressive milieu. Given that tumor-associated macrophages are the predominant immune components in the GBM TME, the recruitment of other effector cells could enhance the immune response. Unlike T-cell-based immunotherapy, NKC-based immunotherapy for cancers evidently avoids graft-versus-host disease [22,23]. This characteristic enables the targeting of tumors without precisely defined antigens for specific responses, offering the potential use of allogeneic products in advance.

In this study, we demonstrated the efficacy of NKC administration in an orthotopic GBM xenograft murine model. In a similar context, Maeoka et al. performed direct intracranial infusion of NKCs in an orthotopic xenograft murine model [35]. However, our approach differed in terms of the route and timing of NKC injection. Although the common route of intravenous injection is via the lateral tail vein, we injected NKCs via the retro-orbital sinus, with the major concern being their delivery to intracranial tumors. Christina et al. reported that there were no significant differences in drug delivery or efficacy between the two procedures. They concluded that the retro-orbital sinus is a safe and effective site for injection [36]. Moreover, according to some studies, immune cells access the brain via the choroid plexus and circumventricular organs, which represent fenestrated capillaries without the blood–brain barrier [37,38]. It is well known that patients with GBM exhibit tumor regions with both disrupted and intact blood–brain barrier [39]. These findings suggest that intravenous administration of NKCs can effectively treat intracranial tumors such as GBM. In fact, Lee et al. reported that the treatment effects of NKCs, which were intravenously injected with increased NKC/tumor-cell ratio, were significantly potentiated compared with intratumoral NKC injection [40]. Moreover, intravenous injection is advantageous for patients ineligible for open surgery, as intratumoral injection of NKCs may not be applicable to them. In clinical settings, several patients cannot undergo surgery owing to tumor location or systemic status, making intravenous administration of NKCs an attractive strategy for GBM treatment. We injected NKCs on day 1 in the intravenous once-infusion group and on days 1 and 7 in the intravenous twice-infusion group, with no significant difference in overall survival between these groups. This indicates that the timing of the second injection was too early to make any substantial difference from a single injection. Administering the second injection later might improve overall survival. Altogether, we found that both the intravenous once- and twice-infusion groups exhibited prolonged overall survival, indicating that intravenous NKC injection effectively controlled GBM formation and growth.

In gene expression analysis, we identified differentially expressed genes between intracranial tumors treated with NKCs and non-treated intracranial tumors. Among the aforementioned GSC and ECM markers, NKC-treated tumors exhibited more downregulated genes (MYC, CD44, STAT3, FN1, and COL6A1), indicating the destruction of primary tumors by NKC activity. In terms of NKC-activating receptor ligands, the expression of CLEC2B and CD70 was downregulated in NKC-treated tumors. CD70 expression, in particular, was downregulated in NKC-treated tumors (fold change: −5.64). CD70 is a ligand of CD27, a member of the tumor necrosis factor receptor family, which is expressed on T cells, B cells, and NKCs [41,42]. Takeda et al. suggested that CD27-mediated activation is related to NKC-mediated innate immunity against cells expressing CD70 [41]. The downregulated expression of CD70 indirectly implies that intracranial tumors attempt to evade NKC attacks during their formation, considering the presence of existing NKCs in the central-nervous-system environment. Further, in terms of NKC inhibitory receptor ligands, the expression of CDH2 was upregulated. CDH2 is a ligand of KLRG1 [19]. Lou et al. recently reported that circulating tumor cells, capable of surviving in the circulation and returning to primary tumors through a self-seeding process, exhibit mechanisms to escape NKC-mediated immune surveillance, with elevated CDH2 expression playing a crucial role [43]. In this escape mechanism, the KLRG1–CD2 axis is important, and targeting N-cadherin seems to be an effective strategy to prevent circulating tumor cells from homing onto the primary tumor and resisting NKC-mediated lysis. In this study, the upregulated expression of CDH2 may indicate a mechanism via which intracranial tumors escape NKC attacks.

As indicated by enrichment analysis, some gene sets associated with various cellular components were downregulated in the NKC-treated group. The major components of the postsynaptic specialization membrane include neurotransmitter receptors and proteins that spatially and functionally organize themselves, such as anchoring and scaffolding molecules, signaling enzymes, and cytoskeletal components [44]. We also detected that glutamate-receptor activity was downregulated in NKC-treated tumors. Some studies have reported the involvement of glutamatergic mechanisms in both glioma progression and glioma-associated epilepsy. Pharmacological intervention against these mechanisms is considered a promising strategy to control tumor progression and epilepsy [45,46,47,48]. The downregulation of gene sets related to glutamate-receptor activity would indicates that the progression of NKC-treated intracranial tumors is regulated by the initial NKC attack. These enriched terms provide insights into the characteristics of tumors after NKC treatment and the key pathways involved in NKC-based immunotherapy.

Summarizing differentially expressed genes between the external and internal layers of intracranial tumors, we observed that the expression of the ECM markers SNED1, LTBP1, COL4A6, and CDH1 was upregulated in the external layer. In a three-dimensional spheroid model, the external layer comprises proliferative cells, the intermediate layer is constituted of quiescence cells, and the inner acidic and hypoxic layer includes necrotic cells [49]. The elevated expression of the aforementioned ECM markers in the external layer of intracranial tumors indicates the upregulation of cell–cell signaling, suggesting an anticancer therapeutic-resistance profile [50].

Despite the strengths of our study, it has some limitations. First, we used NKCs derived from a healthy donor. While allogeneic NKCs do not cause graft-versus-host disease, autologous NKCs are more suitable for clinical applications. However, patients with GBM are in a systemic immunosuppressive environment, and the expansion rate of autologous NKCs poses challenges. We previously reported an efficient feeder-free and chemically defined expansion strategy [51] that could resolve this issue. Second, we used a GBM cell line for our orthotopic xenograft model, which may not completely mimic the GBM TME. While patient-derived GBM cells would be useful for producing a more clinically relevant GBM TME, this approach is technically cumbersome and expensive. Therefore, using GBM cell lines is suitable for widely investigating the potential of novel immunotherapy. Third, we used NOG mice, which are immunodeficient; consequently, we could not assess the dynamics and interaction between administered NKCs and other immune components. To address this limitation, using wild-type mice would be ideal, but GBM formation is difficult in their brain. Investigating the efficacy of NKCs against patients with GBM remains the simplest and most effective method for understanding the dynamics and interaction.

## 4. Material and Methods

### 4.1. GBM Cell Lines

We employed two standard human GBM cell lines: T98G (RIKEN BioResource Research Center, Tsukuba, Japan) and U87MG (American Type Culture Collection, Manassas, VA, USA). These cells were maintained in Dulbecco’s modified Eagle’s medium (Life Technologies, Carlsbad, CA, USA) containing 10% heat-inactivated fetal bovine serum (MP Biomedicals, Tokyo, Japan), 100 U/mL penicillin, and 100 µg/mL streptomycin (Thermo Fisher Scientific, Waltham, MA, USA) at 37 °C in a humidified 5% CO_2_-containing atmosphere.

### 4.2. Induction of NKCs

The expansion of highly purified NKCs was performed using our previously described method [25]. Briefly, peripheral blood mononuclear cells were obtained from 16 mL heparinized peripheral blood from two healthy volunteers (31- and 41-year-old men). The CD3 fraction of peripheral blood mononuclear cells was depleted by RosetteSep^TM^ Human CD3 Depletion Cocktail (STEMCELL Technologies, Vancouver, Canada). These CD3-depleted peripheral blood mononuclear cells were then placed in a T25 culture flask (Corning, Steuben, NY, USA) containing AIM V medium (Life Technologies) supplemented with 10% autologous plasma, 50 ng/mL recombinant human IL-18 (rhIL-18, Medical & Biological Laboratories Co., Ltd., Nagoya, Japan), and 3000 IU/mL rhIL-2 (Novartis, Basel, Switzerland) and cultured at 37 °C in a humidified 5%-CO_2_-containing atmosphere for 14 days. The AIM V medium containing 3000 IU/mL rhIL-2 was replenished, as necessary. 

### 4.3. Animals

We purchased 6–8-week-old female NOG mice from the Central Institute for Experimental Animals (Kanagawa, Japan). All animal experiments were approved by the Institutional Animal Care and Use Committee of Nara Medical University (approval #13403) and conducted in accordance with the Health Guide for the Care and Use of Laboratory Animals and the ARRIVE guidelines for Reporting Animal Research [47].

### 4.4. Growth Inhibition Assays

Growth inhibition assays were performed with U87MG and T98G cells using the xCELLigence Real-Time Cell Analysis (RTCA) DP Instrument (ACEA Biosciences, San Diego, CA, USA), as previously described [48]. Briefly, complete medium (100 μL) was added to each well of an E-plate 16 (ACEA Biosciences), and background impedance was measured at 37 °C in a humidified 5%-CO_2_-containing atmosphere. T98G or U87MG cells (2 × 10^4^/well) were seeded in each well as target cells, and impedance was measured every 5 min for 72 h. After 24 h, genome-edited NKCs were added to each well as effector cells in the predefined effector-to-target cell ratios. Data were analyzed using RTCA v1.2 (ACEA Biosciences).

### 4.5. Orthotopic GBM Xenograft Model and In Vivo Antitumor Activities of NKCs

The in vivo xenograft assay was performed as previously described [49,50]. Briefly, anesthetized NOG mice were secured on a rodent stereotactic frame (SR-6M-HT, Tokyo, Japan). A small drill hole was made at 2 mm right and 1 mm anterior from the bregma, and then 2 µL native Hank’s balanced salt solution containing 105 U87MG cells was infused into the right thalamus at a depth of 3 mm via the drill hole using a Hamilton syringe (33 S-gauge needle) mounted on an infusion syringe pump (Harvard Apparatus, Holliston, MA, USA). The injection was performed over 5 min, left in place for 3 min, and removed over 5 min. Six mice were randomly assigned to the negative background (PBS/IL-2 10,000 IU/mL), six to the intravenous once-infusion (106 NKCs and 10,000 IU/mL IL-2 once), and six to the intravenous twice-infusion (106 NKCs and 10,000 IU/mL IL-2 twice) groups. NKCs or PBS were injected into the retro-orbital sinus using a 29G needle under isoflurane anesthesia, as previously reported [51]. The administration schedule was as follows: on day 0, stereotactic GBM injection was performed. In the negative-background group, IL-2 was intravenously infused on days 1 and 7. In the intravenous once-infusion group, NKCs were infused on day 1 and IL-2 was infused on day 7. The intravenous twice-infusion group was treated on days 1 and 7.

### 4.6. Histochemical Analysis

Intracranial tumors were fixed with 10% neutral buffered formalin and embedded in paraffin. Subsequently, 5 µm-thick sections were placed on glass slides and stained with hematoxylin and eosin. Photographs were obtained using a BX-710 microscope (KEYENCE, Osaka, Japan) at ×40 and ×200 magnifications.

### 4.7. Gene-Expression and Enrichment Analyses

Intracranial tumors were harvested from dead GBM-injected mice. Total RNA from intracranial tumors in the orthotopic GBM xenograft model was extracted using NucleoSpin RNA (Takara Bio, Shiga, Japan), and these RNA samples was sent to Riken Genesis (Kawasaki, Japan), where gene expression analysis was performed using the Clariom^TM^ S array. Microarray data were deposited into GEO (accession no. GSE 248352). Analysis of all CEL files was conducted using transcriptome analysis console v4.1 (Thermo Fisher Scientific). Gene expression was analyzed using the gene-level Signal Space Transformation Robust Multi-Chip Analysis summarization method [52]. Microarray data were normalized with the robust multiarray average method; a probe set was considered expressed if >50% samples exhibited detection above background (DABG) values below the DABG threshold (*p* < 0.05). The expression status, indicating whether specific mRNA was detectable, was denoted as True (T) or False (F). We focused on genes related to glioma stem cell (GSC) markers (NOTCH2, STAT3, MYC, CD44, CXCR4, ITGA6, PDGFRA, L1CAM, NES, SOX2, MSI1, NANOG, CDH5, POU5F1, PROM1, and FUT4), extracellular matrix (ECM) markers (COL6A1, FN1, LTBP1, COL1A1, MMP16, SNED1, CDH1, LUM, CFTR, COL4A6, LAMA1, and SUSD5), chemokines (CCL1, CCL2, CCL3, CCL3L3, CCL4L2, CCL5, CCL7, CCL8, CCL11, CCL13, CCL14, CCL16, CCL17, CCL18, CCL19, CCL20, CCL21, CCL22, CCL23, CCL24, CCL25, CCL26, CCL27, CCL28, CXCL1, CXCL2, CXCL3, CXCL5, CXCL6, CXCL8, CXCL9, CXCL10, CXCL11, CXCL12, CXCL13, CXCL14, CXCL16, CXCL17, XCL1, XCL2, and CX3CL1), chemokine receptors (CCR1, CCR2, CCR3, CCR4, CCR5, CCR6, CCR7, CCR8, CCR9, CCR10, CX3CR1, CXCR1, CXCR2, CXCR3, CXCR4, CXCR5, CXCR6, and XCR1), NKC-activating receptor ligands (CD70, CFP, CLEC2B, ITGB2, MICA, NCR3LG1, NID1/PDGFD, TNFSF4, and TNFSF9, ligands of CD27, NCR1, KLRF1, ICAM1, KLRK1, NCR3, NCR2, OX40L, and CD137, respectively), and NKC inhibitory receptor ligands (CD274, CDH1/CDH2/CDH4, CEACAM1/HMGB1/LGALS9/PTDSS1, COL17A1, PVR, and HLA-E, ligands of PD-1, killer cell lectin-like receptor G1 (KLRG1), TIM3, LAIR1, TIGIT, CD96, and KLRC1, respectively). 

Gene set analysis was performed using gene set enrichment analysis [53]. The reference gene set was c5.all.v2022.1.Hs.symbols.gmt in the Molecular Signatures Database [54].

### 4.8. Statistical Analysis

Statistical analyses were performed using Prism 8 (GraphPad Software Inc., San Diego, CA, USA). The log-rank test was employed for the statistical analysis of survival time. Values represent mean ± standard deviation, and statistical significance of differences was determined using one- or two-way analysis of variance, followed by Tukey’s test. *p* < 0.05 indicated statistical significance.

## 5. Conclusions

To summarize, we demonstrated the antitumor activity of NKCs cultured using our specific method in an in vivo orthotopic xenograft model of GBM. Our gene expression analysis provided a comprehensive overview of the molecular characteristics of intracranial tumors with or without NKC administration, including insights into differences between the external and internal layers of intracranial tumors. These findings lay the groundwork for future investigations into the potential of NKC-based immunotherapy as a promising strategy to treat patients with GBM.

## Figures and Tables

**Figure 1 ijms-25-02435-f001:**
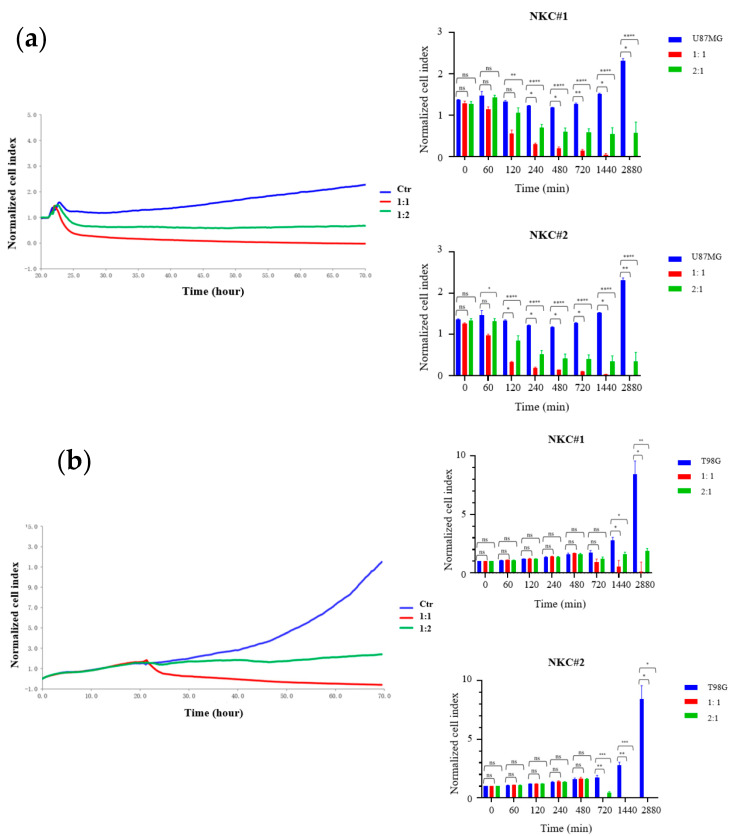
Enhanced growth inhibition of glioblastoma (GBM) cells by natural killer cells (NKCs). The graph on the left shows the growth curves of U87MG (**a**) and T98G cells (**b**) co-cultured with NKCs at effector-to-target cell ratios of 1:1 (red) and 1:2 (green). The blue curve represents cell lines only. The graphs on the right depict real-time cell analysis-based growth inhibition assays. NKC#1 and NKC#2 were derived from another donors. Blue bars represent cell lines only, red bars represent an effector-to-target cell ratio of 1:1, and green bars represent an effector-to-target cell ratio of 1:2. The X and Y axes indicate the co-culture time (min) and relative normalized cell index, respectively. Values represent mean ± standard deviation of 5–6 experiments. Statistical differences were determined by two-way analysis of variance, followed by Tukey’s test. **** *p* < 0.0001, *** *p* < 0.001, ** *p* < 0.01, * *p* < 0.05, ns: not significant.

**Figure 2 ijms-25-02435-f002:**
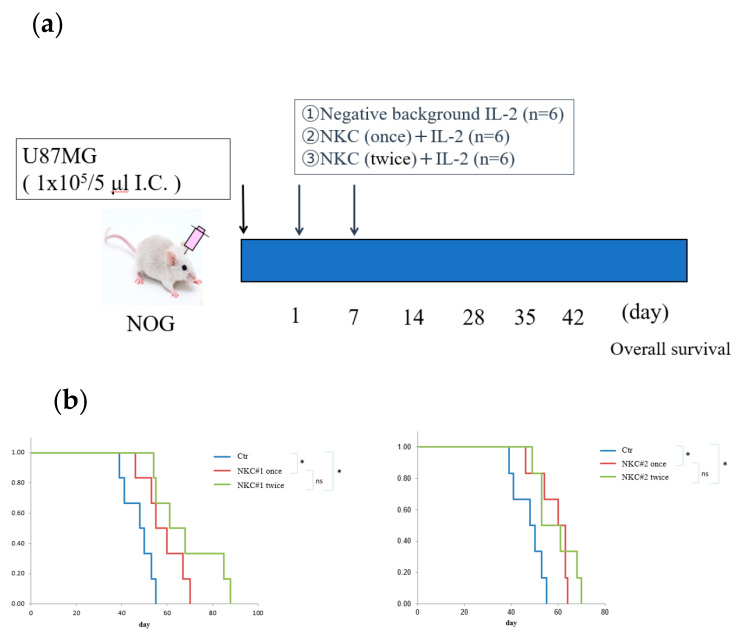
Antitumor effects of natural killer cells (NKCs) in an orthotopic xenograft murine model derived from a glioblastoma (GBM) cell line. (**a**) Schematic of the GBM xenograft model where mice were injected with NKCs via the retro-orbital sinus (*n* = 6/group). (**b**) Kaplan–Meier survival curves for mice treated (once or twice) or non-treated with NKCs. NKC#1 and NKC#2 were derived from another donors. Statistical differences were determined by two-way analysis of variance, followed by Tukey’s test. * *p* < 0.05, ns: not significant. (**c**) Pathological validation of intracranial tumors from each group (negative background, NKC#1, and NKC#2) using hematoxylin and eosin staining. Scale bar, 50 µm.

**Figure 3 ijms-25-02435-f003:**
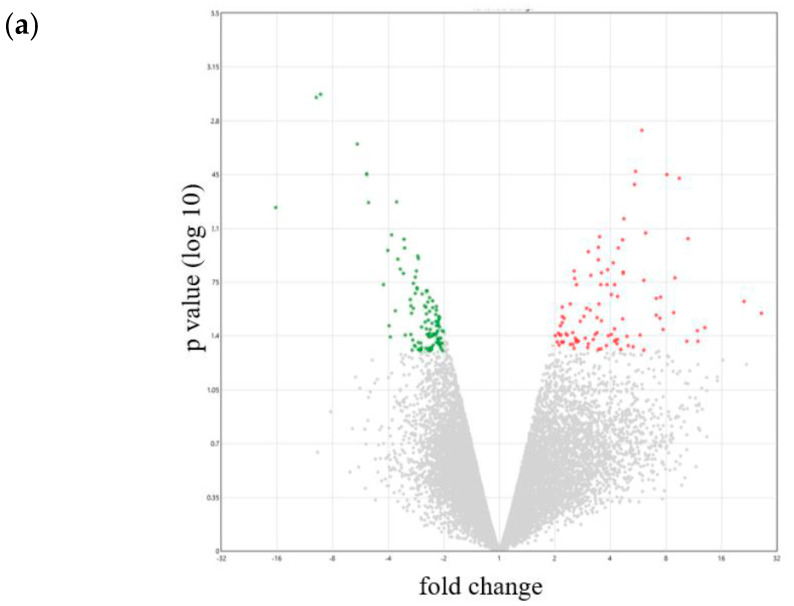
Differential gene expression analysis between natural killer cell (NKC)-treated and non-treated intracranial tumors from the orthotopic glioblastoma (GBM) xenograft model. (**a**) Volcano plot illustrating log_2_-scaled fold change (x-axis) and −log_10_ *p*-value (y-axis) for each gene. (**b**–**g**) Heatmaps of the transcriptome-wide Clariom™ S array of gene expression related to glioma stem cell (GSC) markers (**b**), extracellular matrix (ECM) markers (**c**), chemokines (**d**), chemokine receptors (**e**), NKC-activating receptor ligands (**f**), and NKC inhibitory receptor ligands (**g**). Bar graphs illustrating the normalized enrichment score (NES) (**h**). Enrichment plot depicting downregulated gene sets belonging to different gene ontology (GO) categories (**i**).

**Figure 4 ijms-25-02435-f004:**
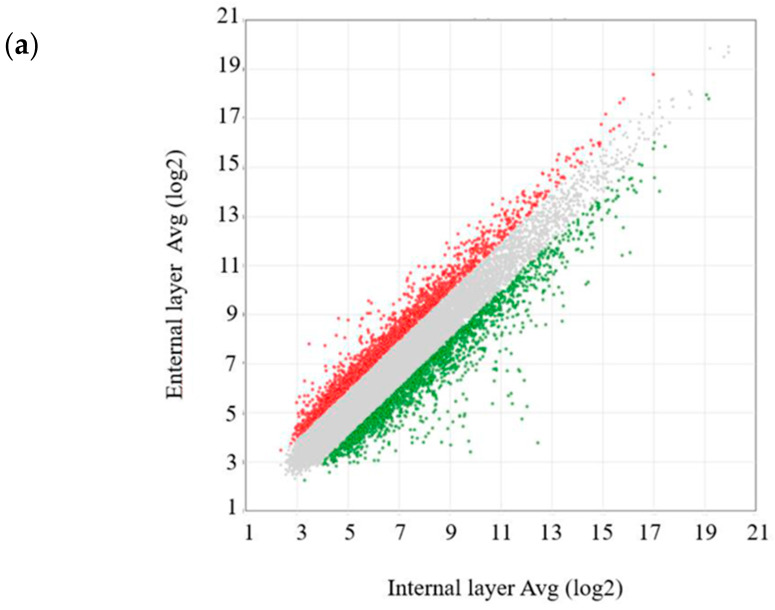
Differential gene expression analysis between the external and internal layers of intracranial tumors from the orthotopic glioblastoma (GBM) xenograft model. (**a**) Volcano plot illustrating log_2_-scaled fold change (x-axis) and −log_10_ *p*-value (y-axis) for each gene. (**b**–**g**) Heatmaps of the transcriptome-wide Clariom™ S array of gene expression related to glioma stem cell (GSC) markers (**b**), extracellular matrix (ECM) markers (**c**), chemokines (**d**), chemokine receptors (**e**), natural killer cell (NKC)-activating receptor ligands (**f**), and NKC inhibitory receptor ligands (**g**).

## Data Availability

Microarray data have been deposited in NCBI GEO. The filtered data for all figures in this study are provided in the Appendix A.

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
