# Peer review of "Antitumor Effects of Intravenous Natural Killer Cell Infusion in an Orthotopic Glioblastoma Xenograft Murine Model and Gene Expression Profile Analysis"

_ijms, 2024, doi:10.3390/ijms25042435_

Round 1

Reviewer 1 Report

Comments and Suggestions for Authors

In this manuscript, the authors investigated the potential of NK cells to fight against GBM by using an orthotopic xenograft model and tested differentially expressed genes. This is a good therapeutic strategy, but it still needs to be supported by more experimental data, and the quality of the writing and presentation of the article needs to be further improved, and I think some revisions need to be completed before publication.

1.  In the intro section, it is necessary to add more information about the mode of effect and characteristics of NK cell therapy for GBM, for example, the introduction of NK cells themselves, how NK cells regulate the tumor microenvironment, antigen presentation to tumor cells after NK cell activation, the infiltration process of T cells, how it specifically affects immune cell subsets of GBM to increase the effect of immunotherapy, how it is possible to enhance the activity of NK cells (cytokines, immune receptors, etc.), the existing clinical therapies based on NK cells, and many others.

2. It is recommended to add in vitro experiments such as testing T-cell activity, testing gene expression based on the results of animal models, and explaining the reasons for the contribution of NK cells. Here are some interesting/important genes, such as CXCL14 and others mentioned in the manuscript.

3. There are some abbreviations in the text, such as ECM, GSC. It is recommended to include their full names for their first appearance.

4. The quality of the figures is not very good, including inconsistent font size and color in the same figure, the presence of other symbols in the axes (Figure 1b), missing bar (Figure 2c), and images that are too large or too small (Figure 3). These are all parts that need to be carefully improved and layout.

5. In the results section, if possible, it is recommended that the authors describe in more detail the results of the experiment, the purpose of utilizing this animal model, what role the genes obtained play in immunotherapy, and more.

6. There are a few grammar mistakes and some very simple mistakes, such as 2.3 (should be 4.3) in line 322 and so on. Authors need to be more careful to double-check the manuscript. 

Reviewer 2 Report

Comments and Suggestions for Authors

Comments to the authors

The authors (Morimoto et al) have Antitumor effects of intravenous natural killer cell infusion in an orthotopic glioblastoma xenograft murine model and gene expression profile analysis. However, following are some of the comments that the authors might find useful for future submission. The manuscript should be revised before publication.

Comment: 

1.      The underlying rationale for the author's preference for Natural Killer Cell (NKC) therapy remains undetermined. It is recommended to provide additional elucidation on both the advantages and disadvantages associated with this therapeutic technique.

2.      To comprehensively address the impact of these treatments on normal cells, the inclusion of investigations involving normal cell lines is imperative. Expanding the discussion to encompass the influence on normal cellular function will contribute to a more thorough understanding of the treatments' effects.

3.      The quality of figures needs to improve.

4.      The language and discussion of the results need improvement.

Comments on the Quality of English Language

Comments to the authors

The authors (Morimoto et al) have Antitumor effects of intravenous natural killer cell infusion in an orthotopic glioblastoma xenograft murine model and gene expression profile analysis. However, following are some of the comments that the authors might find useful for future submission. The manuscript should be revised before publication.

Comment: 

1.      The underlying rationale for the author's preference for Natural Killer Cell (NKC) therapy remains undetermined. It is recommended to provide additional elucidation on both the advantages and disadvantages associated with this therapeutic technique.

2.      To comprehensively address the impact of these treatments on normal cells, the inclusion of investigations involving normal cell lines is imperative. Expanding the discussion to encompass the influence on normal cellular function will contribute to a more thorough understanding of the treatments' effects.

3.      The quality of figures needs to improve.

4.      The language and discussion of the results need improvement.

Round 2

Reviewer 1 Report

Comments and Suggestions for Authors

The authors have responded to comments and revised their manuscript, and I recommend this manuscript for publication.

But there are still some minor format mistakes: "2.4 Growth inhibition assays" in line 336, "2.7 Gene expression and enrichment analyses" in line 368. Be careful and double-check.

Author Response

Thank you for your pointing.

We re-checked the manuscript, and modified the mistakes.